# Spatial inequities in access to medications for treatment of opioid use disorder highlight scarcity of methadone providers under counterfactual scenarios

Eric Tatara[1,2]*, Qinyun Lin[3], Jonathan Ozik[1,2], Marynia Kolak[3], Nicholson Collier[1,2], Dylan Halpern[3], Luc Anselin[3], Harel Dahari[4], Basmattee Boodram[5], John Schneider[6]

1 Consortium for Advanced Science and Engineering, University of Chicago, Chicago, Illinois, United States of America, 2 Decision and Infrastructure Sciences, Argonne National Laboratory, Lemont, Illinois, United States of America, 3 Center for Spatial Data Science, University of Chicago, Chicago, Illinois, United States of America, 4 The Program for Experimental & Theoretical Modeling, Division of Hepatology, Department of Medicine, Stritch School of Medicine, Loyola University Chicago, Maywood, Illinois, United States of America, 5 Division of Community Health Sciences, School of Public Health, University of Illinois at Chicago, Chicago, Illinois, United States of America, 6 Departments of Medicine and Public Health Sciences, University of Chicago, Chicago, Illinois, United States of America

* tatara@anl.gov

**Data Availability Statement:** All data and code used for running experiments, model fitting, and

## Abstract

Access to treatment and medication for opioid use disorder (MOUD) is essential in reducing opioid use and associated behavioral risks, such as syringe sharing among persons who inject drugs (PWID). Syringe sharing among PWID carries high risk of transmission of serious infections such as hepatitis C and HIV. MOUD resources, such as methadone provider clinics, however, are often unavailable to PWID due to barriers like long travel distance to the nearest methadone provider and the required frequency of clinic visits. The goal of this study is to examine the uncertainty in the effects of travel distance in initiating and continuing methadone treatment and how these interact with different spatial distributions of methadone providers to impact co-injection (syringe sharing) risks. A baseline scenario of spatial access was established using the existing locations of methadone providers in a geographical area of metropolitan Chicago, Illinois, USA. Next, different counterfactual scenarios redistributed the locations of methadone providers in this geographic area according to the densities of both the general adult population and according to the PWID population per zip code. We define different reasonable methadone access assumptions as the combinations of short, medium, and long travel distance preferences combined with three urban/suburban travel distance preference. Our modeling results show that when there is a low travel distance preference for accessing methadone providers, distributing providers near areas that have the greatest need (defined by density of PWID) is best at reducing syringe sharing behaviors. However, this strategy also decreases access across suburban locales, posing even greater difficulty in regions with fewer transit options and providers. As such, without an adequate number of providers to give equitable coverage across the region, spatial distribution cannot be optimized to provide equitable access to all PWID. Our study has important

plotting are available on our GitHub repository at https://github.com/hepcep/hepcep_model.

**Funding:** This work is supported by the National Institute on Drug Abuse (NIH) grant U2CDA050098 (J.S., L.A.), by the National Institute of General Medical Sciences grant R01GM121600 (B.B., H.D., J.O.), by the National Institute of Allergy and Infectious Diseases (NIH) grant R01AI158666 (B.B., H.D., J.O.), by the National Institute on Drug Abuse (NIH) grant R01DA043484 (B.B.), and by the U.S. Department of Energy under contract number DE-AC02-06CH11357. The research presented in this paper is that of the authors and does not necessarily reflect the position or policy of the National Institute on Drug Abuse or any other organization. The funders had no role in study design, data collection and analysis, decision to publish, or preparation of the manuscript.

**Competing interests:** The authors have declared that no competing interests exist.

implications for increasing interest in methadone as a resurgent treatment for MOUD in the United States and for guiding policy toward improving access to MOUD among PWID.

## Author summary

Persons who inject drugs (PWID) such as heroin or other illicit substances are at increased risk of getting or transmitting HIV or hepatitis C (HCV) infection, because limited access to clean syringes and drug injection equipment means that PWID often share contaminated syringes. Public health interventions aimed at reducing injection drug use include medications for opioid use disorder (MOUD) that help to reduce cravings for opioids like heroin and fentanyl. However, resources like MOUD provider clinics are often unavailable to PWID due to factors such as the travel distance from home to clinic, frequent clinic visits, along with social phenomena such as degree of medical mistrust and stigma towards treatment. We use a computational simulation model of PWID social networks and drug use behaviors in Chicago, IL and suburbs to show how the geographic distribution of clinical providers of methadone, an MOUD, impacts the injection frequency of PWID. Our modeling results show that when PWID prefer a low travel distance to methadone providers, distributing providers near areas that have the greatest need is best at reducing syringe sharing behaviors. However, this strategy also decreases access across suburban locales, posing even greater difficulty in regions with fewer transit options and providers.

## Introduction

Access to treatment and medication for opioid use disorder (MOUD) is essential in reducing behavioral risks for HIV and HCV infection and overdose associated with injection drug use [1–3]. In addition to individual competing priorities (e.g., unstable housing, childcare), barriers to access to MOUDs among people who inject drugs (PWID) may include structural factors (e.g., drug use-related stigma [4], long travel distances, or policy barriers [5]). Historical, socioeconomic, racial, and other structural factors influence both availability and perception of MOUDs [5–7]. Furthermore, there is a high degree of variability in individual MOUD pharmacology, delivery, and patient preference. As a golden standard to address MOUD access inequities, MOUDs should be available in all communities to facilitate treatment individualization and treatment support retention. As such, understanding access to MOUDs, health services, and other harm reduction services (e.g., syringe service programs) is critical to defining risk environment landscapes that affect fatal and nonfatal overdoses and HIV and HCV infections related to injection drug use.

Effective prevention and treatment strategies exist for opioid use disorder (OUD) but are highly underutilized in the United States. Indeed, only a small fraction (11%) who need MOUD received it in 2020 [8]. Methadone, a synthetic opioid agonist that eliminates withdrawal symptoms and relieves drug cravings by acting on opioid receptors in the brain, is the medication with the longest history of use for OUD treatment, having been used since 1947. A large number of studies support methadone's effectiveness at reducing opioid use [9], but have also shown methadone access disparities along racial, ethnic socio-demographics, and geographic location. While expedient access to methadone maintenance treatment is critical to preventing overdose death [10], at this time its provisioning is restricted to federally licensed

opioid treatment program (OTP) locations, which tend to reflect carceral approaches to treatment such as strict patient surveillance, limited flexibility in medication schedules, and high frequency of travel to OTP locations [5,11]. While access to other MOUDs such as buprenorphine has increased across the US due in part to fewer administrative restrictions, people with OUD deserve options for treatment and many patients prefer methadone even with the geographic access barriers: a factor that motivates the current analysis [12]. This has led to a resurgence in efforts to remove administrative restrictions on clinics and providers to provide methadone to treat OUD among people who use drugs [5].

The goal of this study is to examine the uncertainty in the effects of travel distance in initiating and continuing methadone treatment for OUD and how these uncertainties interact with different spatial distributions of methadone providers to impact co-injection (syringe sharing) risks among PWID. Behavioral risk mitigation (i.e., reduction in syringe sharing) is often a function of the complex interplay of historical, sociological, and structural factors, resulting in nuanced patterns that reflect underlying social and spatial inequities. Research on access to primary health services often cite a preference for a less than 30-minute travel time for individuals seeking care [13,14], though a recent survey on driving times to OTPs showed that almost 18% of the US population would have driving times in excess of 30 minutes to the nearest OTP, and almost 37% of individuals in rural counties experiencing OTP drive times over an hour [15]. Currently, more than a third of continental U.S. zip codes are more than an hour away from treatment, and access to methadone providers remains worse than other MOUD types (ex. buprenorphine and naltrexone) [16]. Providing transportation services has been shown to improve treatment retention for methadone maintenance programs and outpatient drug-free programs [17] and transportation costs have been shown to be a significant factor in travel to OTPs [15]. Minoritized racial/ethnic status has been associated with admission delays for outpatient methadone treatment [6] and reduced likelihood of being offered pharmacologic support for recovery [18]–though once engaged in treatment, have similar retention rates to the majority of clients [19]. Nonetheless, there exists substantial inequities between minority populations' induction, adherence, and treatment completion rates when compared to the White population [20]. At the same time, minoritized groups may prefer accessing treatment services within primary care settings versus specialty mental health clinics [21]. While geographic access to treatment is crucial, access is a multidimensional concept that can be deconstructed into the components of availability, accommodation, affordability, and acceptability [22]. For persons with OUD, access is especially complex because of the interplay of MOUD resource scarcity and drug use stigma, as patients experiencing stigma from MOUD providers are less likely to return [23].

Measuring spatial access to public resources like OTPs must consider the frequency of resource utilization and the mode of transit to the resource [24]. Travel hardships, including extended distances, longer travel, and interstate commute, have been considered as the most common accessibility barriers for people who seek care from distant providers, especially for persons in rural areas where public transportation is limited [15,25–28]. Most existing studies focus on *actual* distance to MOUD locations and very few have studied what is the *ideal* distance (or travel time) preferences to ensure accessibility. The effect of travel hardships on accessibility is most critical for methadone considering the need for daily dosing. Increased distance to treatment can impede daily attendance as shown in a recent study of patients receiving MOUD treatment that found patients residing 10 miles from the treatment facility were more likely to miss doses compared to those who lived within 5 miles [29]. Conventional interventional trials test whether modification of spatial factors is needed, but often difficult and costly to implement.

There is a need for more rapid approaches to assessing and translating spatial epidemiologic findings to practical real-world interventions that benefit proven, yet underutilized interventions such as methadone treatment for MOUD. We examine the impact of methadone provider distribution on syringe sharing among PWID from Chicago, IL, USA and the surrounding suburbs using a validated agent-based model (ABM) (Hepatitis C Elimination in Persons Who Inject Drugs or HepCEP) [2]. Our modeling approach accounts for uncertainties in how individuals perceive access to methadone providers and how that perception affects their decisions to initiate and adhere to MOUD treatment. As such, we employ a robust decision making perspective [30] to capture the effects of different methadone provider distribution approaches across these uncertainties.

## Methods

### HepCEP model

The current study extends our previous work on simulating the PWID population in Chicago and the surrounding suburbs, Illinois, USA, including drug use and syringe sharing behaviors, and associated infection dynamics [2,31]. The demographic, behavioral, and social characteristics of the PWID population is generated using data from five empirical datasets on metropolitan Chicago (urban and suburban) area PWID that is previously described [31]. In brief, this includes data from a large syringe service program (SSP) enrollees (n = 6,000, 2006–13) [32], the IDU data collection cycles of the National HIV Behavioral Surveillance (NHBS) survey from 2009 (n = 545) [33] and 2012 (n = 209) [34], and a social network and geography study of young (ages 18–30) PWID (n = 164) [35]. Of note, profiles of PWID from these data sources were similar when grouped by age, gender and racial/ethnic groups. Data analyses from these sources is used to generate attributes for each of the estimated 32,000 PWID in the synthetic population for metropolitan Chicago [36] in the model and includes: age, age of initiation into injection drug use, gender, race/ethnicity, zip code of residence, HCV infection status, drug sharing network degree, parameters for daily injection and syringe sharing rates, and harm reduction/syringe service program (SSP) enrollment [31]. PWID agents may leave the population due to age-dependent death or drug use cessation and are replaced with new PWID sampled from the input data set to maintain a nearly constant population size of 32,000 for the entire course of the simulation.

Syringe sharing among PWID is modeled in HepCEP via dynamic syringe sharing networks. Network formation is determined by the probability of two PWID encountering each other in their neighborhood of residence and within the outdoor drug market areas in Chicago that attracts both urban and non-urban PWID for drug purchasing and utilization of SSPs that are also located in the same areas [37]. The methods used to calculate network encounter rates, establishment processes, and removal of networks have been described previously [31]. Each modeled individual has an estimated number of in-network PWID partners who give syringes to the individual and out-network PWID partners who receive syringes from the individual. The network edge direction determines the flow of contaminated syringes between individuals, and thus the direction of disease transmission. The network evolves over time, and during the course of the simulation some connections (ties) may be lost, while new ties form, resulting in an approximately constant network size.

MOUD treatment enrollment is modeled in two steps. First, there is an unbiased awareness of MOUD resources by PWID, capturing the knowledge that agents possess about the existence of a methadone provider. The annual target awareness rate, defined as the total annual awareness as a fraction of the total population, is a model parameter with a constant value of 90%. Thus, over the course of a year, 90% of the PWID population will be made aware of

MOUD treatment and, subsequently decide whether to engage in MOUD treatment. The total PWID target MOUD treatment awareness for a single day is determined by the daily mean treatment awareness, which is the total PWID population multiplied by the annual treatment awareness parameter / 365. The daily awareness target is sampled from a Poisson distribution using the daily mean treatment awareness. The model assumes that all PWID who wish to obtain MOUD treatment are able to do so, and there are no constraints on individual provider treatment capacity.

PWID that receive MOUD treatment experience a reduction in the number of daily drug injections, which is determined by multiplying the PWID's baseline pre-MOUD daily drug injection frequency by a reduction multiplier sampled from a uniform distribution from 0 to 0.25 [38]. Thus, the mean reduction in daily injection frequency is 87% when on MOUD treatment compared to when not on MOUD. Reduction in daily injection frequency reduces the number of syringe sharing episodes with other infected individuals.

## Reasonable geographic access assumptions

Our approach to model the initiation and continuation of MOUD treatment incorporates multiple aspects of access to care: 1) the travel distance to the nearest methadone provider from the PWID place of residence, 2) the frequency of clinic visits, and 3) racial/ethnic inequalities to treatment as represented by the geospatial heterogeneity of the PWID population demographics. PWID first decide to enroll in MOUD treatment and then subsequently decide to continue treatment every 7 days, a duration chosen to reflect the average frequency of clinic visits for the treatment over time. Average treatment duration for methadone is obtained from literature to be 150 days [39] and the probability to start and continue treatment every 7 days is estimated using a random empirical distribution with mean of 150 days, resulting in some PWID having longer or shorter time on treatment. PWID who discontinue methadone treatment may reinitiate treatment later during the course of the simulation. Different urban and non-urban travel distance preferences to the nearest methadone provider are used by PWID to determine if they will or will not enroll in MOUD treatment (Table 1). The probability that a PWID will enroll in MOUD treatment is greater when the treatment travel distance is below the travel distance preference.

We define six different possible combinations of low, medium, and high travel distance preferences considering three urban/suburban distance preference combinations for each low, medium, high travel distance preference pair with a maximum travel distance preference, and three corresponding low, medium, and high travel distance preferences pairs with no maximum distance preference (Table 1). In the cases with maximum preferences, the individuals will not be able to access treatment if the provider is farther away than the maximum preferred travel distance.

**Table 1. Travel distance to methadone provider preferences used for reasonable geographic access assumptions.** The table represents six different possible combinations of low, medium, and high travel distance preferences considering three urban/suburban distance preference combinations for each low, medium, high distance preference pair with a maximum distance limit, and three corresponding low, medium, and high distance preference pairs with no maximum distance limit.

| Travel Distance Preference | Travel Distance to Methadone Provider (miles) | | | | | |
|---|---|---|---|---|---|---|
| | With Maximum Distance Limit | | | No Maximum Distance Limit | | |
| | Urban | Suburban | | Urban | Suburban | |
| Low | 1 (max: 15) | 5 (max: 60) | | 1 | 5 | |
| Medium | 2 (max: 15) | 10 (max: 60) | | 2 | 10 | |
| High | 5 (max: 15) | 20 (max: 60) | | 5 | 20 | |

To approximate reasonable geographic access, the travel distance in miles from zip code centroid to the nearest methadone provider is calculated using the *sf* package in R (version 4.0.2) [40] which provides a Euclidean distance metric. Travel network routing is not modeled in this analysis. Methadone treatment requires frequency of visits comparable to that of people's grocery shopping (daily or weekly) [41,42], and travel distances of 1 mile (urban) and 10 miles (suburban/rural) areas is reasonable for community members, respectively, to their grocery stores. Because of the scarcity of methadone providers, we extend the urban reasonable travel distance preference to 2 miles for urban areas. Published findings indicate that access to mental health treatment within 10 miles is associated with greater attendance in persons with OUD [43]. Accordingly, the travel distance preference of "reasonable geographic access" to methadone providers is set at 2 miles in urban areas, which approximates a 30-minute walking distance, and for suburban and rural areas, the travel distance preference is set at 10 miles (Table 1, Medium travel distance preference). We define Low and High travel distance preferences to further account for uncertainties around minimal and maximal travel distance preference.

As there is limited information on how geographic access affects individuals' decision to seek treatment, an additional layer of uncertainty is introduced via penalties on individuals' probability of getting treatment when the geographic travel distance exceeds the reasonable access distance preference but is below the maximum distance limit. If the travel distance exceeds the preference, the per-decision probability of treatment is lower (by a factor θ) than if the PWID is closer to the provider. Since θ is not easily estimated, values are chosen ranging from 60% to 90% for this study, which represents broad per-decision penalties for accessing locations beyond distance preferences. The base probabilities are calculated by using the distribution of the PWID agent population under the actual spatial distribution of methadone providers (see below) and under different preference scenarios to match the overall methadone treatment duration values. Each penalty level is combined with each of the six travel distance preference combinations in Table 1, resulting in 18 separate parameter combinations, or reasonable access assumptions.

## Spatial distribution of methadone providers

**MOUD provider data.**   This study includes data on methadone maintenance MOUD providers in Chicago and the surrounding suburbs, which we define as the 298 zip codes from Cook County (i.e., the most populous county in Illinois and includes Chicago) and the five collar counties that border Cook, which are also the next five most populous counties in the state. We include providers in zip codes beyond the Chicago metropolitan area for the state of Illinois (1,085 additional zip codes) to provide context in interpretation, though only perform simulations and evaluate scenarios within these boundaries. Zip code and population data are obtained from the 2010 U.S. Census [44]. Spatial boundary and clinical data used in the analysis can be found in the Opioid Environment Policy Scan Data Warehouse, version 1.0 [45]. A total of 81 Illinois providers are identified by specifying "methadone maintenance"from the Substance Abuse and Mental Health Service Administration (SAMHSA) Behavioral Health Treatment Service Locator (derived from the 2019 National Survey of Substance Abuse Treatment Service) [46].

**Counterfactual methadone provider spatial distributions.**   To study how the spatial distribution of methadone providers affects syringe sharing behaviors among PWID, three counterfactual distributions are generated to spatially redistribute methadone provider locations. That is, the geographic locations of methadone providers are changed and re-evaluated for accessibility to those providers. In all scenarios, the total number of all methadone providers in Illinois is assumed to be constant (n = 81).

*Spatially random*: MOUD treatment locations are randomly distributed within the study area (298 zip codes in Cook and five collar counties) and other areas (1085 zip codes) in Illinois. The total number of modelled methadone providers in Illinois remains unchanged; only the location of these resources changes. This distribution provides a useful null hypothesis of spatial randomness that can be benchmarked against actual geographic distribution of resources, as well as alternate counterfactual distributions.

*Need-based 1*: Methadone provider locations are assigned proportionally to the adult population (age 18–39) within each zip code which results in more methadone providers assigned to zip codes with larger adult populations. The Hamilton (largest remainder) method [47] is used to calculate the number of methadone providers assigned to each zip code and to ensure that each area was assigned an integer number of methadone providers. Specifically, methadone providers are first allocated to each zip code proportional to the local at-risk population. The result for each zip code consists of an integer part plus a fractional remainder, or in some cases, only a fractional remainder. Each zip code is first allocated an integer number of providers. This leaves some providers unallocated. The zip codes are then ranked based on the fractional remainders: one additional methadone provider is added to the zip code areas ranking the highest until all providers are allocated.

*Need-based 2*: In this distribution, methadone providers are assigned proportionally to the total PWID population [31] for each zip code. The difference between *Need-based 1* and *Need-based 2* is how the need for methadone within each zip code area is estimated. Need-based 2 is potentially better reflects local geographic needs as the PWID population likely represents a closer approximation for the need for methadone providers than an area's entire adult population.

**Outcome: Syringe sharing.** The reduction in annual syringe sharing events among PWID who are adherent to methadone treatment relative to a baseline scenario without methadone availability is investigated as the main outcome of interest of the simulation studies. Syringe sharing reduction is calculated for each of the 18 reasonable access assumptions in each of the three counterfactual methadone provider distributions, along with the actual provider distribution. A baseline simulation is first conducted to determine the number of annual syringe sharing events in each zip code when PWID have no awareness of methadone providers, and do not enroll in MOUD treatment. The baseline is not sensitive to provider distribution or travel distance preference since no MOUD treatment occurs.

For each combination of reasonable access assumption and provider distribution, the syringe sharing reduction metric is defined as the difference in the number of annual syringe sharing events in each zip code when PWID are aware of methadone providers, relative to the baseline. The HepCEP model is run for a 20-year period starting in 2010 through simulated year 2030. The total number of syringe sharing events in each zip code is tabulated only for year 2030, resulting in the metrics for annual syringe sharing reduction. The simulation time frame is based on the need to initialize the model using population data calibrated to year 2010, and to allow the model population and network dynamics to stabilize, as has been done in prior studies [22,24].

A total of 1,440 simulations were conducted using high-performance computing workflows implemented with the EMEWS framework [48]. The 1,440 runs include 20 stochastic replicates for each of the 72 parameter sets, where each parameter set corresponds to the four provider spatial distributions for each of the 18 reasonable access assumptions. We report the mean number of syringe sharing episodes across the 20 stochastic replicates. The simulation experiments were executed on the Bebop cluster run by the Laboratory Computing Resource Center at Argonne National Laboratory.

Annual syringe sharing reduction relative to baseline (no MOUD) is aggregated across all zip codes to produce a single scalar metric for each of the 72 parameter sets. We define a decision regret score to represent the difference in syringe sharing reduction for each of the four spatial provider distributions, relative to the spatial distribution with the largest reduction in syringe sharing, for each combination of reasonable access assumptions. A decision regret score of zero represents the best outcome in terms of reducing syringe across each of the four spatial provider distributions, for a specific combination of reasonable access assumptions. Conversely, a high regret score means that the scenario had a significantly larger number of syringe sharing episodes relative to the best scenario with the fewest number of syringe sharing episodes.

The 75th-percentile of the regret score distribution for each of the four provider spatial distribution scenarios is used to evaluate the robustness for each spatial strategies, i.e., adequate performance over a wide range of possible ground truths and decision-making uncertainties.

## Results

### Spatial access to methadone providers under different reasonable geographic access assumptions

Fig 1 provides a geographic illustration of whether each zip code minimum travel distance to a methadone provider is within the travel distance preference of reasonable geographic access (Table 1), underlying different assumptions of what is the ideal distance to ensure accessibility. The first, second, and third row in Fig 1 corresponds to the low, middle, and high travel distance preferences, respectively in Table 1. For each travel distance preference (each row in Fig 1), the four figures (columns in Fig 1) show each zip code's accessibility to the nearest methadone provider under the actual spatial distribution of providers and three counterfactual spatial distributions.

Comparing the actual provider spatial distribution with the two need-based distributions (Fig 1, column-wise), we identify areas where the need for methadone providers is high while the spatial access to providers is limited. For example, some areas in Chicago have high need but few providers, and accessibility to methadone providers are improved in the two need-based counterfactual distributions.

Comparing across rows in Fig 1, more zip codes have better spatial access to methadone providers as we assume a higher travel distance preference (i.e., people are able and willing to travel longer distances). This preference does not account for transit barriers such as travel time and access to public transit infrastructures, vehicles, as well as the financial cost of transit. Under the low travel distance preference that likely reflects real-world barriers to transit, few areas have reasonable geographical access to methadone providers. Therefore, simply redistributing methadone providers spatially may not provide better access when the number of individual providers is limited.

### Effects of spatial distribution of methadone providers on annual syringe sharing reduction by zip code

Since the total number of methadone providers are fixed in this analysis, redistributing provider locations in the counterfactual distributions relative to the actual distribution leads to some areas having a higher reduction in syringe sharing events than others. The reduction in the number of annual syringe sharing events (relative to baseline) in each zip code for the actual provider distribution (Fig 2, columns 1 and 5), under each reasonable access assumption reflect the zip code provider spatial accessibility in Fig 1. The change in syringe sharing

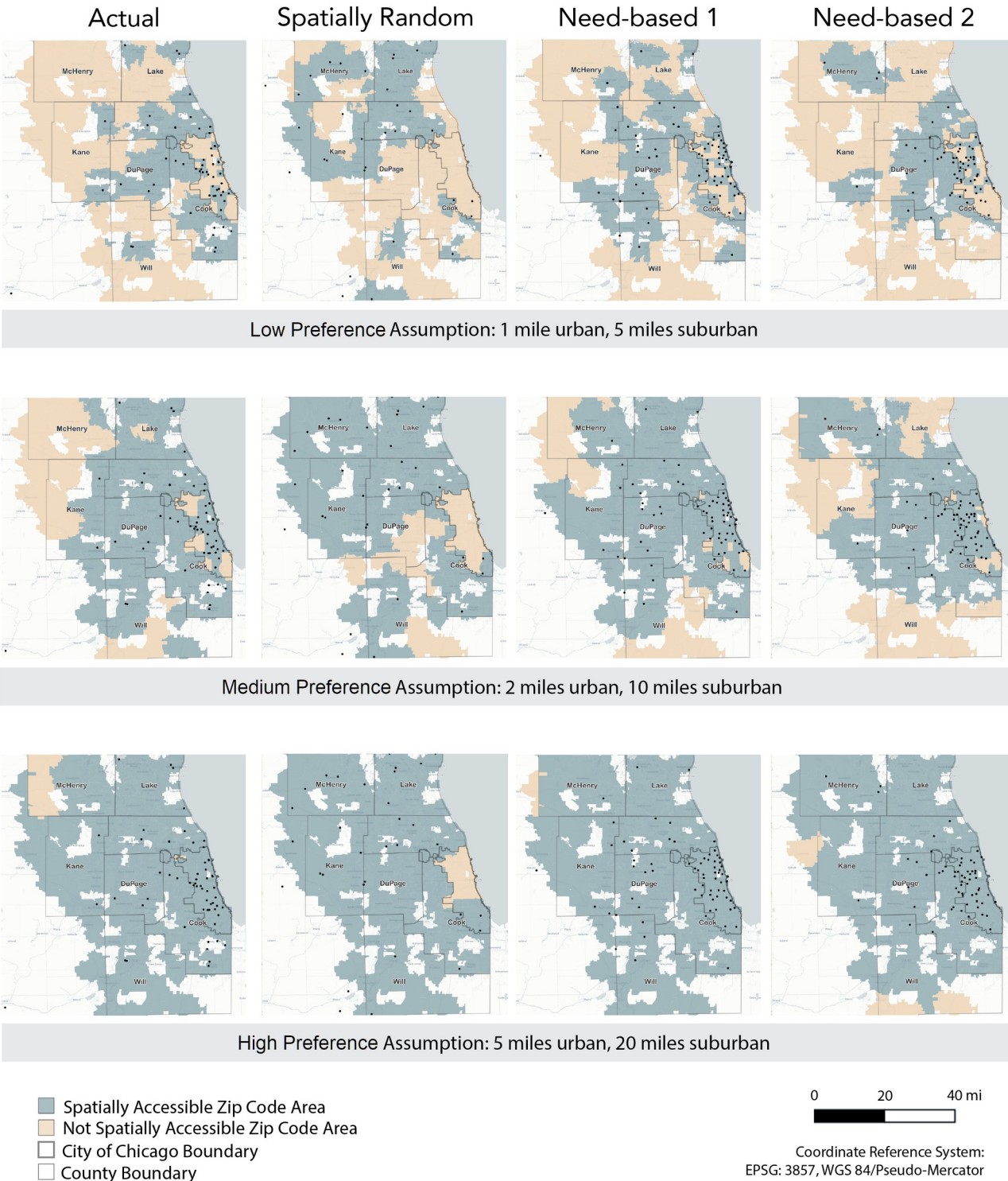

**Fig 1. Spatial access to methadone providers for the actual scenario and three counterfactual distribution scenarios, under varying travel distance preference assumptions.** Each dot represents the location of a single methadone provider. City of Chicago and collar county borders are indicated. Spatial access to methadone providers is calculated as distance to nearest provider to the center of each zip code area; thus in the low travel distance preference assumption, zip codes areas are not identified as accessible if there is no provider within a mile of its geographic center. Esri. "Light Gray Canvas" [basemap]. Scale Not Given. "Light Gray Canvas Base". Oct 26, 2017. https://www.arcgis.com/home/item.html?id=291da5eab3a0412593b66d384379f89f. (October 1, 2023).

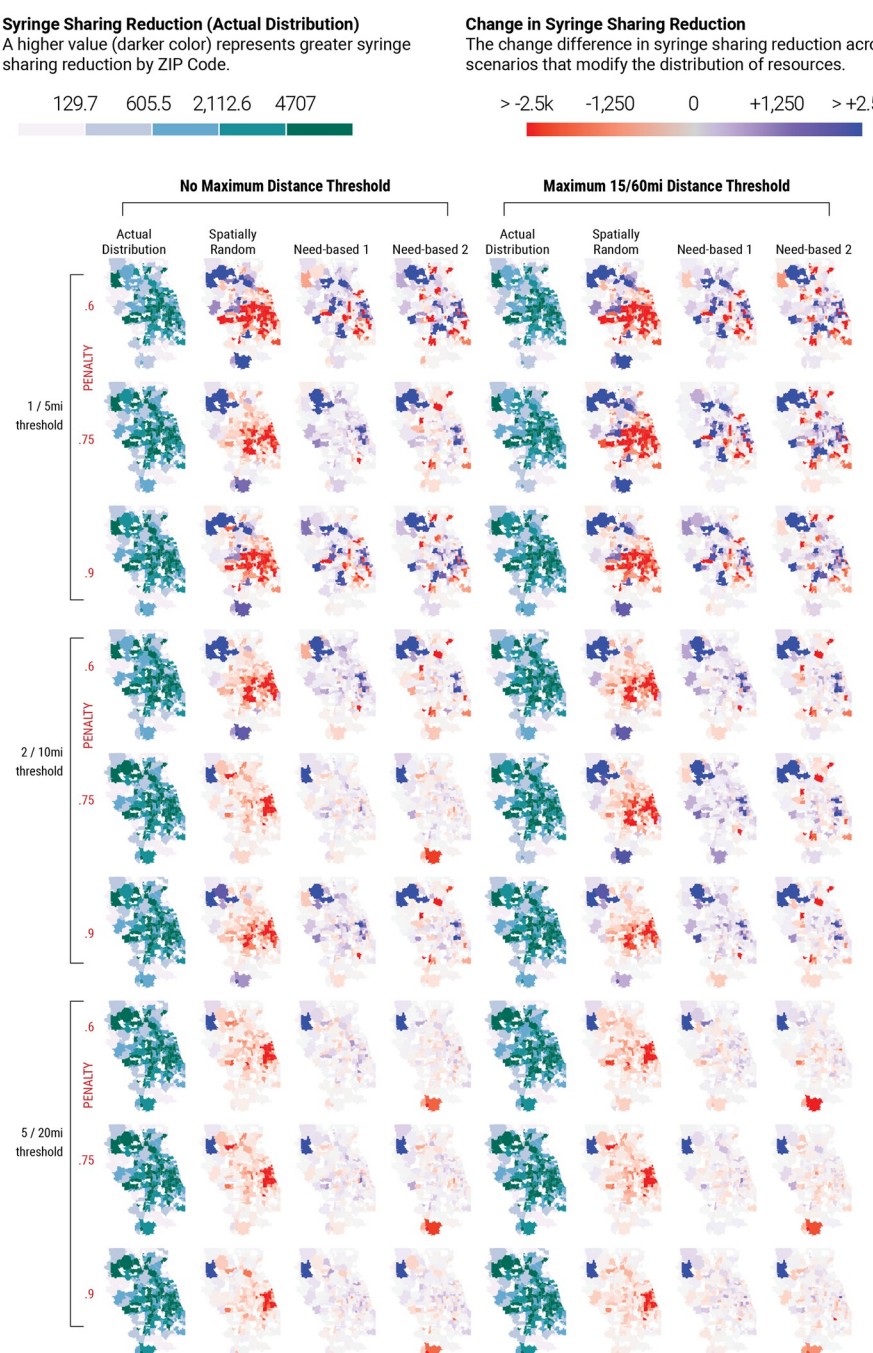

**Fig 2. Effects of reasonable access to methadone assumptions on syringe sharing events by zip code for each scenario in Cook County IL and surrounding counties.** Columns 1 and 5 represent the reduction in syringe sharing events for the actual spatial distribution of methadone providers, relative to the baseline scenario without methadone. In the other columns, blue colored zip codes indicate a greater reduction in syringe sharing events relative to the actual scenario, while red colored zip codes indicate a lesser reduction in syringe sharing events relative to the actual scenario. Zip code polygons shapefile obtained from the 2010 U.S. Census Datasets [44].

reduction by zip code, relative to the actual distribution, highlights the effects of spatially redistributing methadone providers in each of the three counterfactual distributions (Fig 2, columns 2–3 and 6–8). Blue colored zip codes indicate a larger reduction in annual syringe

sharing events than the actual distribution, while red colored zip codes indicate a lesser reduction in syringe sharing than the actual distribution.

Simulation scenarios across a range of methadone provider location distributions and spanning the spectrum from least optimistic reasonable access assumptions (Fig 2, upper rows, low travel distance preference, maximum travel distance, and a penalty of 0.6.) to the most optimistic access assumptions (Fig 2, lower rows, high travel distance preference, no travel distance maximum, and a penalty of 0.9) demonstrate a high degree of spatial heterogeneity in the expected reduction in syringe sharing events among PWID in Chicago, IL and surrounding suburbs.

## Robustness of spatial methadone provider distributions in reducing annual syringe sharing events

The spatial variation of syringe sharing reduction across reasonable access assumptions (Fig 2) reflects heterogeneity in the PWID population both in terms of local population density, and in terms of drug use behaviors and co-injection risks reflected in the underling empirical population data used in the model. Since the underlying individual PWID behaviors are difficult to observe in reality, optimizing provider spatial distributions to reduce the number of syringe sharing events may not be an appropriate goal; rather, the robustness of spatial distributions that perform well over a wide range of possible ground truths and decision-making uncertainties are examined.

Fig 3 provides a visual representation of the regret scores for the annual reduction in syringe sharing events for each of the 18 reasonable access assumptions, grouped by the four spatial distributions of methadone providers (Tables A and B in S1 Supporting Information). Each point in Fig 3 represents the regret score for each of the reasonable access assumptions, color coded by travel distance preference (Table 1). Given the definition of regret scores, lower values represent a more ideal outcome (less regret) for a particular reasonable access assumption, i.e., values of zero indicate that the spatial provider distribution had the largest reduction in syringe sharing events for the indicated reasonable access assumption. Vertical box plots for each spatial provider distribution in Fig 3 provide the median, 25th and 75th percentiles for regret score.

Fig 3 therefore helps to provide insight as to how reasonable access assumptions impact individuals' decisions to initiate and continue methadone treatment. The need-based 2 distribution (PWID density) performs better than the need-based 1 (total population density) and actual distributions when spatial access to providers is important (i.e., low travel distance preference and higher barriers to travel, represented as orange dots). The need-based 1 distribution performs better than the need-based 2 and actual distributions under medium and high travel distance preference (e.g. willingness to travel further and lesser barriers to travel, represented as blue and green dots).

Across the actual, spatially random, and need-based 1 provider distributions, the regret scores tend to rank from highest for the low travel distance preferences (Fig 3, orange dots), with the next highest for the medium travel distance preference (Fig 3, blue dots), and lowest for the high travel distance preference (green dots) due to the fact that more PWID chose to initiate and continue MOUD treatment when travel distance preferences are greater and therefore allow for more access to providers based on the locations shown in Fig 1. Low travel distance preference access assumptions result in the lowest regret for the need-based 2 distribution because providers are located closest to areas of highest PWID density, resulting in better access compared to other distributions.

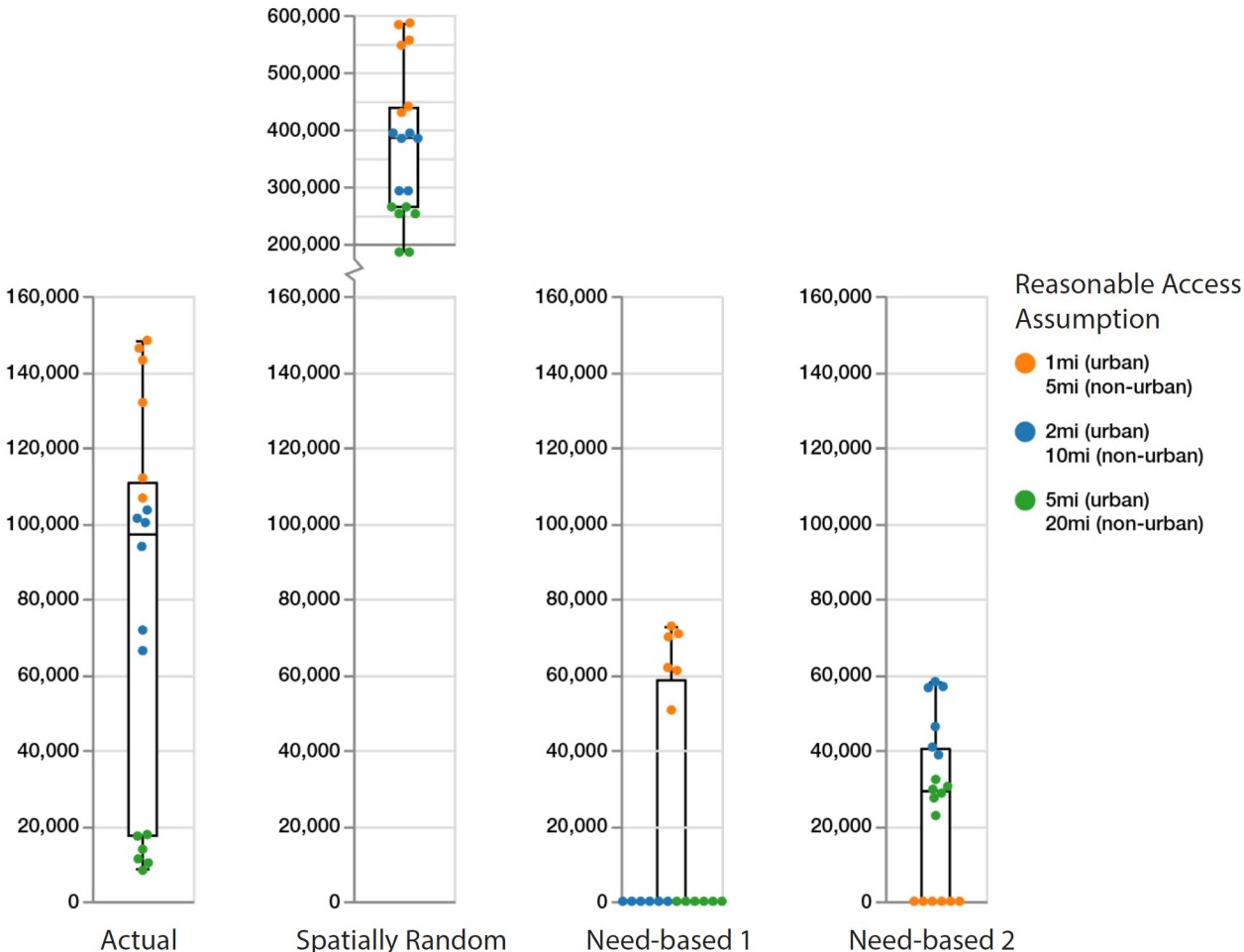

**Fig 3. Regret score of annual reduction in needle sharing by travel distance preference and spatial distribution of methadone providers.** Each dot represents the regret score for each of the 18 reasonable access assumptions (three travel distance preferences with and without maximum distance plus three distance penalty values). Vertical box plots for each spatial provider distribution provide the median, 25th and 75th percentiles. Whisker lines indicate minimum and maximum regret values. The regret score represents the difference in syringe sharing reduction for each of the four spatial provider distributions, relative to the spatial distribution with the largest reduction in syringe sharing, for all combinations of reasonable access assumptions. A decision regret score of zero represents an ideal outcome in that the spatial distribution was best at reducing syringe sharing for a given reasonable access assumption. Conversely, a high regret score means that the scenario had a significantly larger number of syringe sharing episodes relative to the best scenario with the fewest number of syringe sharing episodes.

The actual provider distribution results in a greater reduction in syringe sharing events than the need-based 2 distribution (PWID density) only when assuming a high travel distance preference (green dots), while it performs much worse than both need-based distributions when assuming low travel distance preference (orange dots). Notably, the actual provider distribution does not achieve a zero (best) regret score for any combination of reasonable access assumptions (Fig 3). In all cases, the spatially random distribution generates the worst result (Fig 3).

Based on a 75th percentile regret metric for each of the four spatial provider distributions, the Need-based 2 spatial distribution (PWID density) represents a more robust distribution of methadone providers with respect to reducing annual syringe sharing events, across the uncertainties around all reasonable access assumptions and travel distance preferences (Fig 3, fourth column).

## Discussion

Our agent-based modeling study of PWID from Chicago and the surrounding Illinois suburbs provides valuable insights into the development of future interventions to enhance MOUD treatment uptake by PWID. We found that the impact of the spatial distribution of methadone providers on syringe sharing frequency is dependent on assumptions of access. When there is a low travel distance preference for accessing methadone providers, i.e., PWID are faced with significant structural barriers, distributing providers near areas that have the greatest need (defined by density of PWID) is optimal (Fig 3). However, this strategy also decreases access across suburban locales, posing even greater difficulty in regions with fewer transit options and providers (Fig 2). As such, without an adequate number of providers to give equitable coverage across the region, spatial redistribution cannot be optimized to provide equitable access to all persons (and potential persons) with OUD. Policies that would expand geographic access to methadone maintenance treatment by making it available at pharmacies and/or federally qualified health centers may better meet the need of this population [11,15,49], and are currently an area of vigorous debate and consideration [5].

The PWID population in Chicago and the surrounding suburbs [50] and other urban areas [36] is well-characterized. Detailed and current knowledge on PWID demographics can be used to study how access to MOUD treatment providers can be improved over existing resource distributions, along with estimates of future needs due to shifts in PWID demographics and locations. While each context is uniquely different, some of the core metrics that impact treatment availability, namely density of providers and travel to clinic, can be applied in rural, urban and other diverse contexts. Future work should expand the model to other locales to ensure that assumptions around factors that drive access to care are consistent, and how any heterogeneity might be explained.

For all reasonable access assumptions and provider location distributions, spatially redistributing methadone providers relative to the actual distribution may effectively decrease access in some areas. There were no scenarios that exhibited zero areas with worse access compared to the actual scenario, highlighting the scarcity of providers in the region as a major challenge. Geospatial visualization of our simulation results (Figs 1 and 2) show that the more remote and less populated areas remained inaccessible, reflecting urban-suburban accessibility challenges. Underserved areas could be supplemented with mobile treatment providers to target these vulnerable populations.

Under modeling scenarios with substantial uncertainties as in the current study, particularly related to underlying individual behaviors that are difficult to observe, optimizing spatial provider distributions to reduce syringe sharing among PWID may not be an appropriate goal. Instead, robust [30] methadone provider location distributions that perform well over a wide range of possible ground truths and uncertainties should be sought. Detailed, data-driven, agent-based models combined with the capacity for large-scale computational experimentation, can provide such analyses to support decision making under uncertainties, or when empirical data collection is costly or unethical. Our results show that the Need-based 2 spatial distribution (PWID density) represents the most robust distribution of methadone providers with respect to reducing annual syringe sharing events, across the uncertainties around all reasonable access assumptions and travel distance preferences (Fig 3, fourth column).

Need-based counterfactuals were more like the actual provider distribution than the spatially random distribution, suggesting that some areas' needs for methadone providers are being met. However, some geographic locales remain in high need of providers, as demonstrated by the need-based scenarios (Figs 1 and 2). McHenry County, in the northeastern part of the study area, is notable for having all or most of its zip codes characterized by no access in

all travel distance preference assumptions–despite a large PWID population in need of MOUD treatment options. Many nearby suburban counties likewise have a patchwork of access across travel distance preference assumptions. While some regions of Chicago have access to providers, more access on transit-connected northern and lake coastal sides of the city would better support populations who currently need, or may need, treatment.

The low travel distance preference assumption highlights multiple, significant gaps in access across the Chicago area and surrounding suburban counties. While this assumption may seem restrictive, it may also be the most realistic. For example, 1- and 5-miles traveled in urban or suburban areas for a resource required daily or weekly is considered exceptionally reasonable in food access literature (where grocery stores may also be accessed weekly). This low travel distance preference assumption may also be optimistic because of additional social, economic, and structural barriers faced at opioid treatment programs providing methadone services, like cost and drug use stigma (experienced at the provider and/or neighborhood that it is located within). Our study has important implications for guiding policy toward improving access to MOUD among PWID, particularly in areas where the population is dispersed, e.g., expansive suburban areas in large metropolitan cities like Chicago.

## Limitations

Our current results report reductions in annual syringe sharing events for all combinations of reasonable access assumption and provider spatial distributions. Downstream health sequelae such as hepatitis C and HIV have been examined in previous work [2,51] in the PWID population; however, the current study did not investigate associated reductions in HCV. The current study also did not implement HCV treatment or other harm reduction services (e.g., sterile syringe and equipment provision). Simply reducing the syringe sharing frequency in a highly connected PWID network may not be sufficient to eliminate new HCV infection without also reducing the disease incidence.

The reported results include the annual reduction in syringe sharing for only a single simulation year (2030). Time-varying trends in syringe sharing metrics were not investigated. Further, the PWID population is maintained at a constant size of 32,000 individuals for the course of the simulation. Although we model transient changes in PWID demographics as in previous studies [2,31], we believe that the PWID population size may be somewhat close to constant given that people who transition to MOUDs is balanced out by new initiates into injection drug use entering the population.

The model has several limitations around practical access to methadone treatment. First, laws around access to methadone and OTP services can vary greatly by state, with some states like Illinois requiring a government issued ID for access to services [52]. The HepCEP model does not account for inequity in access to OTPs based on an individual's possession of a valid government issued ID card. Minoritized groups, especially those who are unhoused, recently released from carceral settings, or undocumented, may face additional barriers to obtaining methadone treatment for OUD due to lack of government issued ID documents. Second, the model assumes that there is no treatment capacity constraint on individual providers, e.g. every PWID who seeks methadone treatment is able to do so. OTPs across the US have experienced constraints in providing care due in part to governmental regulations and lack of coverage via private health insurance for methadone treatment [53]. In light of this, our results may overestimate access to methadone treatment based on the study locale.

Synthetic model population data used in this analysis predominantly represent minoritized community members. The authors acknowledge that including social phenomena such as degree of medical mistrust, racism and stigma towards treatment are all important factors in

urban MOUD induction and adherence [54], but are beyond the scope of this analysis. These unmeasured, potentially impactful forces may further moderate the findings described.

Travel distance and time to providers was calculated using Euclidean distance between PWID's residence zip centroid and the nearest methadone provider location. The HepCEP model does not model travel network routing of PWID between residence and methadone provider via roads or sidewalks, which would account for longer travel distance. An explicit model of travel network routing would need to account for available modes of transportation (car, rail, bus, walking) to each PWID and is beyond the scope of this modeling effort. However, the use of the travel distance penalties described in the Reasonable Geographic Access Assumptions Section does account for uncertainties around individuals' probability of obtaining treatment based on travel distance thresholds.

## Supporting information

**S1 Supporting Information.** Table A. Overall syringe sharing reduction and regret scores under for each reasonable access assumption under each scenario. Table B. Overall syringe sharing reduction relative risk and regret scores for each reasonable access assumption under each scenario.
(DOCX)

## Acknowledgments

This work was completed with resources provided by the Laboratory Computing Resource Center at Argonne National Laboratory (Bebop cluster).

## Author Contributions

**Conceptualization:** Jonathan Ozik, Marynia Kolak, Nicholson Collier, Basmattee Boodram.

**Data curation:** Luc Anselin, Basmattee Boodram.

**Formal analysis:** Eric Tatara, Qinyun Lin, Jonathan Ozik, Marynia Kolak, Dylan Halpern.

**Funding acquisition:** Jonathan Ozik, Harel Dahari, Basmattee Boodram, John Schneider.

**Investigation:** Eric Tatara, Jonathan Ozik, Marynia Kolak, John Schneider.

**Methodology:** Eric Tatara, Qinyun Lin, Jonathan Ozik, Nicholson Collier, Harel Dahari, Basmattee Boodram.

**Project administration:** Jonathan Ozik, John Schneider.

**Resources:** Jonathan Ozik.

**Software:** Eric Tatara, Nicholson Collier, Dylan Halpern, Luc Anselin.

**Supervision:** Jonathan Ozik.

**Visualization:** Qinyun Lin, Dylan Halpern.

**Writing – original draft:** Eric Tatara, Qinyun Lin, Jonathan Ozik, Marynia Kolak, Nicholson Collier, Harel Dahari, Basmattee Boodram, John Schneider.

**Writing – review & editing:** Eric Tatara, Qinyun Lin, Jonathan Ozik, Marynia Kolak, Nicholson Collier, Harel Dahari, Basmattee Boodram, John Schneider.

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
