## [Decision Letter · Decision Letter 0]

1 Apr 2024

Dear Dr. Tatara,

Thank you very much for submitting your manuscript "Spatial inequities in access to medications for treatment of opioid use disorder highlight scarcity of methadone providers under counterfactual scenarios" for consideration at PLOS Computational Biology. As with all papers reviewed by the journal, your manuscript was reviewed by members of the editorial board and by several independent reviewers. The reviewers appreciated the attention to an important topic. Based on the reviews, we are likely to accept this manuscript for publication, providing that you modify the manuscript according to the review recommendations.

The Authors are expected to address all the criticisms by all Reviewers. In particular, the authors clarify and assess the use of Euclidean distance on the findings (Reviewers #1 & #2), clarify the source of zip code and spatial data (Reviewer #1), address the substantial friction between people (particularly minority population groups) and treatment utilization, discuss the challenge on PWID data availability (Reviewer #2). In additional to the above comments, please address,

1. Figure 3, could the authors provide an explanation why the regret score showed a very different relation between travel distance preference and regret score for Need-based 2 compare to the other three scenario? (e.g. highest regret score 2mi urban / 10mi non-urban)

2. Also, could the authors comment on whether mid-distance preference (2mi urban / 10mi non-urban) having the highest regret score is a general result for Need-based 2, or is a finding due to the distribution of the population/PWID in Chicago?

3. Could the authors comment on the generalizability of the study findings, such as its applicability in other cities in the US, or in less urbanized areas?

4. Following on Reviewer #2’s comment, if PWID data is limited or unavailable, what would be the best strategy?

Sincerely,

Eric HY Lau, Ph.D.

Academic Editor

PLOS Computational Biology

Thomas Leitner

Section Editor

PLOS Computational Biology

The Authors are expected to address all the criticisms by all Reviewers. In particular, the authors clarify and assess the use of Euclidean distance on the findings (Reviewers #1 & #2), clarify the source of zip code and spatial data (Reviewer #1), address the substantial friction between people (particularly minority population groups) and treatment utilization, discuss the challenge on PWID data availability (Reviewer #2). In additional to the above comments, please address,

1. Figure 3, could the authors provide an explanation why the regret score showed a very different relation between travel distance preference and regret score for Need-based 2 compare to the other three scenario? (e.g. highest regret score 2mi urban / 10mi non-urban)

2. Also, could the authors comment on whether mid-distance preference (2mi urban / 10mi non-urban) having the highest regret score is a general result for Need-based 2, or is a finding due to the distribution of the population/PWID in Chicago?

3. Could the authors comment on the generalizability of the study findings, such as its applicability in other cities in the US, or in less urbanized areas?

4. Following on Reviewer #2’s comment, if PWID data is limited or unavailable, what would be the best strategy?

Reviewer's Responses to Questions

**Comments to the Authors:**

Reviewer #1: The manuscript focuses on modeling MOUD access among PWID individuals. This is a very important topic, and I applaud the authors for focusing their research efforts on this area. The agent-based modeling approach looks solid, but there are a few questions that need to be addressed:

- It is not clear where the zip code and spatial data came from. For example, in 2010, per the US Census, Illinois had 1,384 zip code tabulation areas vs the 1085 “zip codes” noted in the paper. https://www.census.gov/geographies/reference-files/2010/geo/state-local-geo-guides-2010/illinois.html

- The paper notes that “Most existing studies focus on actual distance to MOUD locations, and very few have studied what is the ideal distance (or travel time) preferences to ensure accessibility.”. While preference is indeed important to model, so is accurate modeling of the actual distances. This is the biggest shortcoming I noticed within this analysis. The R package sf results in Euclidean distance between two lat/long locations and not a routable distance. Many of the modeled locations end up being in Chicago, which is split by multiple rivers and interstates. Novaes and Valente, Love et al, and others have shown that routable distances vs Euclidean distances can vary from 1.2 to 1.4 (typically), depending on rural/urban and other factors. An “adjustment” was stated to have been made in urban areas from 1 mile to 2 miles, but this adjustment was made solely on the scarcity of methadone clinics.

- Table 1 on page 8 does not seem to show the 2-mile adjustment in urban areas that is then noted on page 9.

- It is not clear whether the modeling accounts for persons who start treatment but drop out. The paper notes “every 7 days” and the average treatment duration to be 150 days. However, did the model implement a normal bell-shaped distribution of treatment duration or peg all the persons modeled to be at 150 days?

- MOUD laws vary significantly from state to state. Illinois is one of only 8 states that require a government ID to take part in an opioid treatment program. Not all PWID persons may fit that or other requirements. The paper should, at a minimum, speak to the wide variety of laws that differ from state to state. From a broader applicability perspective, this is important. https://www.pewtrusts.org/en/research-and-analysis/issue-briefs/2022/09/overview-of-opioid-treatment-program-regulations-by-state

- Additional clarifications are needed from a modeling MOUD distribution perspective about any assumptions that were made as to a location's ability to service a maximum number of individuals. Facilities of all types have their maximum service abilities, and MOUD locations can vary in capacity.

Reviewer #2: The review is uploaded as an attachment.

**Have the authors made all data and (if applicable) computational code underlying the findings in their manuscript fully available?**

Reviewer #1: **No: **Understandably, the residential locations of the persons who used injection drugs could not be provided.

Reviewer #2: None

PLOS authors have the option to publish the peer review history of their article (what does this mean?). If published, this will include your full peer review and any attached files.

Reviewer #1: **Yes: **Dr. Matthew Hudnall

Reviewer #2: **Yes: **Penelope Mitchell

Figure Files:

Data Requirements:

Reproducibility:

References:

---

## [Editor Report · Decision Letter 1]

9 Jul 2024

Dear Dr. Tatara,

We are pleased to inform you that your manuscript 'Spatial inequities in access to medications for treatment of opioid use disorder highlight scarcity of methadone providers under counterfactual scenarios' has been provisionally accepted for publication in PLOS Computational Biology.

Best regards,

Eric HY Lau, Ph.D.

Academic Editor

PLOS Computational Biology

Thomas Leitner

Section Editor

PLOS Computational Biology

Thanks for addressing all the editor’s and reviewers' comments. Congratulations on the excellent work!

---

## [Editor Report · Acceptance letter]

23 Jul 2024

PCOMPBIOL-D-23-01805R1 

Spatial inequities in access to medications for treatment of opioid use disorder highlight scarcity of methadone providers under counterfactual scenarios

Dear Dr Tatara,

I am pleased to inform you that your manuscript has been formally accepted for publication in PLOS Computational Biology. Your manuscript is now with our production department and you will be notified of the publication date in due course.

With kind regards,

Anita Estes
